# Inferring stimulation induced short-term synaptic plasticity dynamics using novel dual optimization algorithm

**Alireza Ghadimi**[1,2,3], **Leon Amadeus Steiner**[1,4,5,6], **Milos R. Popovic**[2,3], **Luka Milosevic**[1,2,3], **Milad Lankarany**[1,2,3,7] *

**1** Krembil Research Institute – University Health Network (UHN), Toronto, Ontario, Canada, **2** Institute of Biomaterials & Biomedical Engineering (IBBME), University of Toronto, Toronto, Ontario, Canada, **3** KITE Research Institute, Toronto Rehabilitation Institute - University Health Network (UHN), Toronto, Ontario, Canada, **4** Department of Neurology, Charité-Universitätsmedizin Berlin, Berlin, Germany, **5** Berlin Institute of Health (BIH), Berlin, Germany, **6** Institute of Neurophysiology, Charité-Universitätsmedizin Berlin, Berlin, Germany, **7** Department of Physiology, University of Toronto, Toronto, Ontario, Canada

* milad.lankarany@uhnresearch.ca

**Data Availability Statement:** Data and Codes will be available through NSBSPL Github.

**Funding:** This study was supported by the following grants and scholarships: Natural

## Abstract

Experimental evidence in both human and animal studies demonstrated that deep brain stimulation (DBS) can induce short-term synaptic plasticity (STP) in the stimulated nucleus. Given that DBS-induced STP may be connected to the therapeutic effects of DBS, we sought to develop a computational predictive model that infers the dynamics of STP in response to DBS at different frequencies. Existing methods for estimating STP–either model-based or model-free approaches–require access to pre-synaptic spiking activity. However, in the context of DBS, extracellular stimulation (e.g. DBS) can be used to elicit presynaptic activations directly. We present a model-based approach that integrates multiple individual frequencies of DBS-like electrical stimulation as pre-synaptic spikes and infers parameters of the Tsodyks-Markram (TM) model from post-synaptic currents of the stimulated nucleus. By distinguishing between the steady-state and transient responses of the TM model, we develop a novel dual optimization algorithm that infers the model parameters in two steps. First, the TM model parameters are calculated by integrating multiple frequencies of stimulation to estimate the steady state response of post-synaptic current through a closed-form analytical solution. The results of this step are utilized as the initial values for the second step in which a non-derivative optimization algorithm is used to track the transient response of the post-synaptic potential across different individual frequencies of stimulation. Moreover, in order to confirm the applicability of the method, we applied our algorithm–as a proof of concept–to empirical data recorded from acute rodent brain slices of the subthalamic nucleus (STN) during DBS-like stimulation to infer dynamics of STP for inhibitory synaptic inputs.

Sciences and Engineering Research Council of Canada, RGPIN-2020-05868 (M.L) AGEWELL, UofT FASE Graduate Student and Postdoctoral Award in Technology and Aging (A.G.); Deutsche Forschungsgemeinschaft (DFG, German Research Fundation) Project ID 424778381 TRR 295 (L.A. S.); Junior Clinician Scientist Program of the Berlin Institute of Health (L.A.S.); German Academic Exchange Service, DAAD (L.A.S.); Brain Canada Foundation (L.M.); Walter and Maria Schroeder Foundation (L.M.) M.L. designed the study, A.G. and M.L. analyzed data and developed dual optimization algorithm. L.A.S collected data (related to a previously published study). A.G. and M.L. wrote the manuscript, L.A.S and L.M. contributed in the preparation of the manuscript. A. G., L.A.S, M.R.P, L.M, and M.L. revised and edited the manuscript.

**Competing interests:** The authors have declared that no competing interests exist.

## 1. Introduction

Short-term synaptic plasticity (STP) is an essential property of neuronal networks that is involved in information processing of the brain [1–3]. STP enables neurons to communicate with each other through multiple frequency bands [1]. This property can alter the strength of a synapse based on the history of presynaptic activities and regulates the frequency band of the transmitted information. According to recent experimental observations in humans and animals that receive invasive electrical stimulations [2, 4–6], the strength of the synapse–reflected by the amplitude of the recorded postsynaptic currents in the rat or field potentials in single unit recordings of the human brain–varies based on the frequency of electrical stimulation. Such studies demonstrated that high-frequency electrical stimulation induces short-term synaptic depression in the stimulated nuclei [2, 4, 7, 8]. Additionally, theoretical studies [2, 9–11] utilized different models of STP and showed that synaptic depression during high frequency DBS is the most prominent feature of suppression of firing rates in stimulated nuclei. Rosenbaum et al. [11] showed that axonal and synaptic failures can cause synaptic depression in high frequency DBS. Accordingly, they derived a computational model to reproduce the suppression of β oscillations (13-30Hz) observed in parkinsonian patients during DBS. Alternatively, Farokhniaee et al. [10] and Milosevic et al. [2] used the Tsodyks-Markram (TM) model of STP to incorporate the impact of different types of synaptic plasticity in a leaky integrate and fire model to replicate firing patterns of stimulated neurons. The TM model used in these studies can justify the experimental observation. However, the parameters of stimulation-induced STP were not well characterized in previous theoretical studies. In the present study, we propose a novel parameter estimation technique to estimate the dynamics of stimulation-induced STP by the well-known TM model.

Tsodyks and Markram introduced a phenomenological model that accurately represents the dynamics underlying facilitation and depression observed in short-term synaptic plasticity [12, 13]. Although several other models have been developed to describe dynamics of STP, the TM model and its extended versions [5, 12, 14, 15] have been wildly used due to their simplicity and interpretability of underlying parameters. Generally speaking, one can benefit from techniques in time series analysis to fit nonlinear models like the TM model to recorded neural recordings (i.e., postsynaptic potentials/currents) (see [16, 17]). Thus, the parameters of the TM model can be estimated using least mean square error (LMSE) approaches [12] in which the parameters are yielded to minimize the mean square error (MSE) between the model output (i.e., the postsynaptic current) and the recorded postsynaptic currents. However, the LMSE-based algorithms are prone to local minima and might result in low accuracy parameter estimation [12].

To address the local minima problem, probabilistic approaches are proposed by Costa et al. [12] to provide an estimation of the posterior distribution of the TM parameters and the uncertainty of the estimation [12, 13]. More recently, a new method was developed by Ghanbari et al. [15, 18] to estimate the TM parameters from extracellular recordings by utilizing a generalized linear model (GLM) concurrent to the TM model and reproducing the firing rate of the postsynaptic neuron. Ghanbari et al. [15, 18] also used a different GLM to directly estimate the STP dynamics, a model that can be considered as an alternative to the TM model. Rossbroich et al. [19] introduced a new synaptic model which represents short-term dynamics by combining an exponential kernel with a non-linear readout function. The simplicity of this model enabled applying the concepts of STP in artificial neural networks to examine its role in learning.

Since the dynamics of the TM model or any other STP models are frequency dependent, it is crucial to infer STP parameters across different frequencies of presynaptic activations.

Traditionally, STP parameters were adjusted to fit the model to a neuronal recording during single frequency stimulations or a single frequency stimulation with one recovery spike after one second period of silence [12, 15]. Costa et al. [12] showed that using stochastic poisson point process as the presynaptic stimulation significantly improves the accuracy and the speed of their probabilistic algorithm. Ghanbari et al. [15, 18] also used a poisson process for the pre-synaptic spike firing, which is more efficient in estimating the TM parameters. In another study, by Amidi et al. [20], they developed a statistical method that can estimate the synaptic parameters of a sparse network of neurons given their connectivity and spiking activity which are assumed as stochastic processes. Despite the benefits of using stochastic firing patterns, this type of stimulus is not common in experimental protocols.

Recent studies showed that STP is involved in the underlying mechanism of deep brain stimulation (DBS) of multiple subcortical regions [2, 8, 21, 22]. The experimental procedure for studying STP during DBS involves applying multiple individual frequencies of stimulation. Using only one stimulation frequency cannot provide enough accuracy in estimating all parameters. For instance, in Costa's paper [12] the estimation posterior distribution of facilitation time constant is large, which shows the low accuracy of this strategy. Even if the parameters provide consistent output at the specific frequency that is being used for the inference, they are not reliable to predict the STP for other unobserved stimulation frequencies.

Designing an algorithm for estimating the TM model parameters accurately requires a deep understanding of the dynamics of the postsynaptic current (PSC) in response to DBS-like stimulations. Since with DBS at a fixed frequency the inter-pulse intervals remains unchanged, the PSC reaches a steady-state value after a transient state. Therefore, the DBS-induced PSC response comprises of transient and steady-state components. Fitting the TM model to DBS recordings without considering these components might result algorithms that converge to local minima.

To resolve this issue, we divided the PSCs into transient and steady-state components which capture different features of the postsynaptic response. It is worth mentioning that the steady-state values of the PSC for different stimulation frequencies can be calculated analytically and one can benefit from fast gradient-based optimization algorithms to estimate the TM model parameters. However, we show in this paper that the estimated parameters based on the steady-state values of the PSC cannot necessarily replicate the underlying transient response of the TM model. In other words, the TM model might have similar steady-state values of the PSC, but different transient responses.

We developed a dual optimization algorithm that incorporates the estimated parameters of the TM model calculated from steady-state values of the PSC as initial parameters for an algorithm that further estimates those parameters based on the transient responses of the PSCs. This approach improves the accuracy and speed of the parameter's estimation. Using extensive synthetic data, we demonstrated that the performance of the dual optimization algorithm significantly outperforms that of conventional methods. Moreover, as a proof of concept, we applied the algorithm on experimental recordings of the acute slice of subthalamic nucleus (STN) from a single individual rat in response to stimulation, to infer the synaptic model of the rat STN and show the accuracy of the model by replicating the experimental observations.

## 2. Results

### 2.1. Electrical stimulation pulses as pre-synaptic spikes of a stimulated neuron

Recent experimental studies in the human brain [2, 4] demonstrated that high frequency DBS induces STP in the stimulated nuclei. In-vitro experiments in rat have suggested that high

frequency electrical stimulation of STN induces synaptic depression which can be observed in the recorded postsynaptic currents of the stimulated nuclei. A recent theoretical framework on the cellular mechanisms of DBS [2] revealed that DBS-evoked excitatory and inhibitory neuronal responses (of various substructures of the basal ganglia and thalamus in human brain) are the results of simultaneous activations of convergent afferent inputs. To this end [2], we assumed that the timing of stimulation pulses can be considered as pre-synaptic (stimulation-evoked) spikes (the effect of axonal failure can be adjusted in the model) which simultaneously activate all afferent inputs. Here, we used the TM model to describe stimulation-induced STP. The TM model is a set of differential equations that simulate the dynamics of short-term facilitation (STF), short-term depression (STD), and the postsynaptic current that is generated as a result of release of neurotransmitters in the synaptic cleft. To cover a wider range of synaptic dynamics we utilized an extended version of the TM model as follows.

$$\frac{du}{dt} = \frac{U - u(t)}{F} + f(1 - u(t^-))\delta(t - t_{spk}),\tag{1}$$

$$\frac{dR}{dt} = \frac{1 - R(t)}{D} - u(t^-)R(t^-)\delta(t - t_{spk}),\tag{2}$$

$$\frac{dI}{dt} = -\frac{I}{\tau_{syn}} + Au(t^+)R(t^-)\delta(t - t_{spk})\tag{3}$$

where $u$ represents the neurotransmitters utilization probability that manifests the STF dynamics, and $r$ represents the fraction of available neurotransmitters that mimics STD dynamics. The {$f$, $U$, $F$, $D$} are the parameters of the model denoting magnitude of facilitation, baseline release probability, facilitation time constant, and depression time constant, respectively. The effect of each presynaptic action potential (stimulation-evoked spike in this study) was expressed by the Dirac delta function, firing at $t = t_{spk}$. The arrival of spikes to the synaptic terminal triggers the release of the neurotransmitter vesicles and transfers the neuronal signal to the postsynaptic neuron. The release of neurotransmitter appears on the postsynaptic input current, regulates the probability of the transmitters release and the number of available vesicles (Eqs 1 and 2). Each of these processes will be recovered to their initial values with their specific time constants. Since the time constant of these two recovery processes are not equal, the synaptic efficacy will change for the next spike arriving at the presynaptic terminal. The postsynaptic current decays to zero with neurotransmitter time constant ($\tau_{syn}$), which is determined by the type of neurotransmitter and the synaptic connection (e.g. $\tau_{syn} \approx 3\ ms$ for glutamatergic synapses, and $\tau_{syn} \approx 10ms$ for GABAergic synapses).

In this study, we used excitatory synapses of $\tau_{syn} \approx 3\ ms$ to generate the synthetic data. It is to be noted that our results are valid for both excitatory and inhibitory synapses. Moreover, we neglected the modulation of the synaptic delay as it is very small compared to the time constants of the STP [23].

## 2.2. Steady-state and transient response of TM model to electrical stimulation

To model stimulation-induced STP, we replaced $t_{spk}$ in the TM model with the stimulation events. In consistent with experimental protocols, the inter-spike time intervals of stimulation pulses for each individual frequency of stimulation were kept constant. Thus, we simplified the TM model by only considering the inter-spike interval of stimulation pulses (i.e., a constant number for each individual frequency of stimulation). We referred to this model as Discrete-

time Tsodyks-Markram model. The discrete-time TM can be written as:

$$u[n] = f + (1-f)\left(U + (u[n-1] - U)exp\left(\frac{-1}{F_{DBS} \times F}\right)\right), \tag{4}$$

$$R[n] = 1 + ((1 - u[n-1]) \times R[n-1] - 1)exp\left(\frac{-1}{F_{DBS} \times D}\right) \tag{5}$$

$$I[n] = I[n-1]exp\left(\frac{-1}{F_{DBS} \times \tau_{syn}}\right) + Au[n]R[n] \tag{6}$$

where $F_{DBS}$ denotes the frequency of stimulation. $u[n]$, $r[n]$, and $I[n]$ indicate $u(t_n^+)$, $r(t_n^-)$, $I(t_n^+)$, respectively (as in Eqs 1 and 2), and $t_n$ is the time of the $n$-th stimulation pulse (presynaptic spike). $I[n] = I(t_n^+)$ is the instantaneous postsynaptic response to $n$-th stimulation pulse, reflecting the peak of the postsynaptic current (PSC).

As $n$ increases, the peak of PSC converges to a steady-state value and remains unchanged. This steady-state value can be formulated based on Eqs 4 and 5 of discrete-time TM model as follow:

$$u_\infty = \frac{f + (1-f)Uexp\left(\frac{-1}{F_{DBS}\times F}\right)}{1 - (1-f) \times exp\left(\frac{-1}{F_{DBS}*F}\right)}; \tag{7}$$

$$R_\infty = \frac{1 - exp\left(\frac{-1}{F_{DBS}\times D}\right)}{1 - (1-u_\infty) \times exp\left(\frac{-1}{F_{DBS}\times D}\right)}; \tag{8}$$

$$I_\infty = \frac{u_\infty R_\infty}{1 - exp\left(\frac{-1}{F_{DBS}\times\tau_{syn}}\right)} \tag{9}$$

where $u_\infty$ and $I_\infty$ represent the steady-state values of the utilization probability and the peak of PSC, respectively.

To validate whether the steady-state values calculated by Eq 9 were matched with those obtained by the TM model (or equivalently the discrete-time TM model), we generated the PSC of the TM model with a facilitatory synapse given different frequencies of DBS-like pulses. As it is shown in Fig 1A, the steady-state values of the PSC of both TM and discrete-time TM models fit those obtained by analytical solutions across all stimulation frequencies. Several examples of the time traces of the PSC responses and their analytically calculated steady-state responses (Eq 9) were shown in Fig 1B.

## 2.3. Inferring STP parameters using steady-state values of postsynaptic currents

Since the relationship between the steady-state values of PSCs and TM model parameters can be formulated in an analytical form, one can estimate these parameters by a solving a system of equations given enough data points (i.e., steady-state PSCs for different stimulation frequencies). However, since these equations are not linear and the recordings are noisy, we utilized an optimization algorithm to find a set of parameters that minimizes the objective function defined based on the peak of the PSCs as derived in Eqs 7–9. Here, we used a gradient-based optimization algorithm, including a trust-region method, to find the optimal set of the parameters that minimize the L2 norm of the difference between the steady-state PSC calculated in

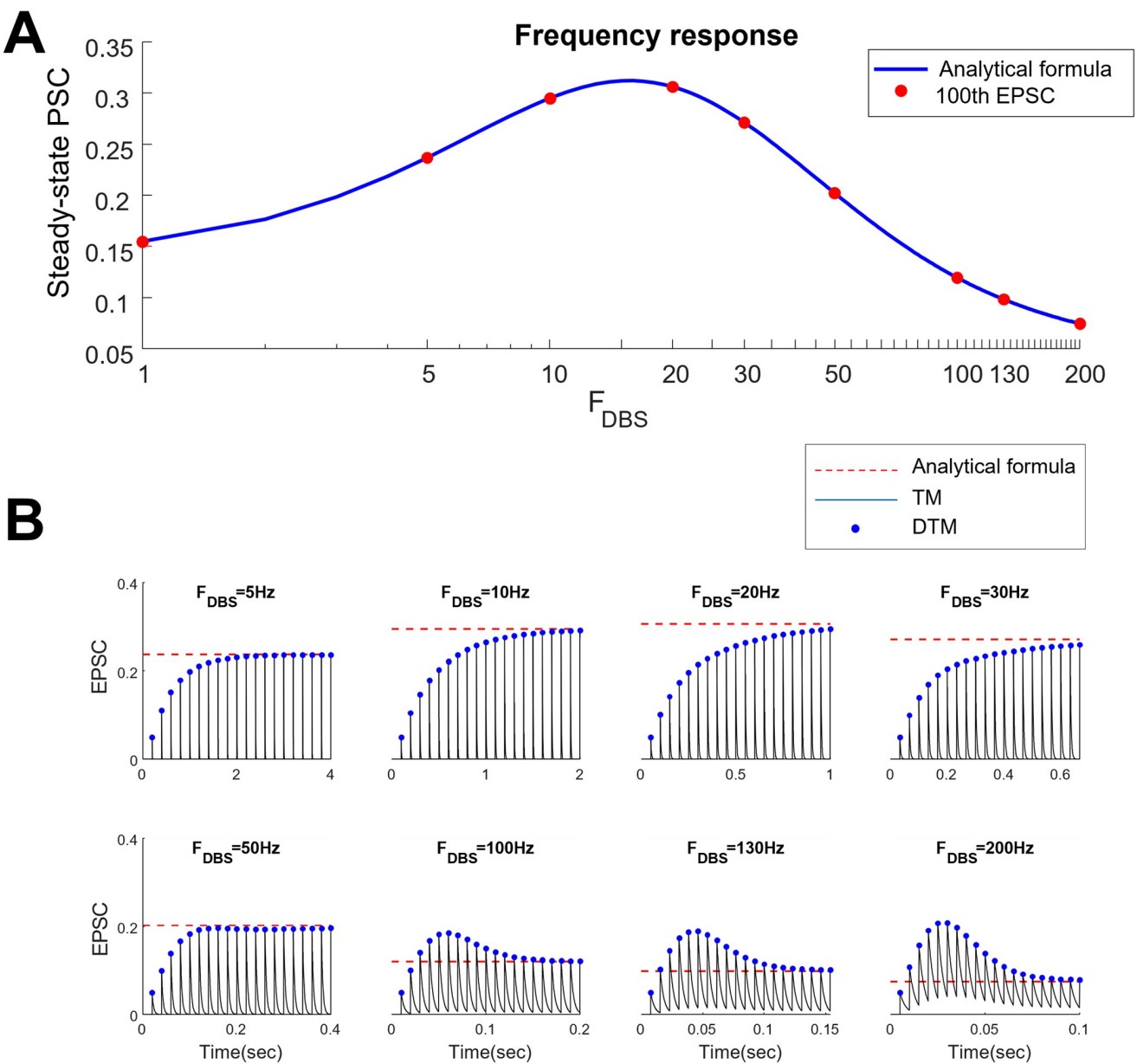

**Fig 1. Steady-state responses of postsynaptic current (PSC) in response to different stimulation frequencies. A)** The steady-state value of PSCs calculated by the analytical formula (blue line) accurately fits to the steady-state responses (red dots) of the discrete TM model (calculated for the 100th stimulation pulse). **B)** An example of PSC in response to 1, 5, 10, 20, 30, 50, 100, 130, 200 Hz stimulation pulses. The red dashed line represents the steady-state value of PSCs calculated from analytical solution (see methods). The black plot shows the time trace of postsynaptic PSC in response to the stimulation calculated from the TM model. The blue dots represent the local peaks of the PSC calculated from discrete TM model. Each panel shows the response of PSC to 30 stimulation pulses at different frequencies. The steady-state values of both TM and discrete TM models converge to those obtained by analytical solution.

Eq 9 and that obtained experimentally across different stimulation frequencies. This cost function can be written as:

$$Steady-sate\ error = \sum_{freq \in F_{DBS}} \left( I_\infty^{freq} - \hat{I}_\infty^{freq}(\theta) \right)^2 \tag{10}$$

where $I_\infty^{f\,req}$ is the steady-state value of PSCs at stimulation frequency of $f\,req$, selected from the set of recorded frequencies, $F_{DBS}$. $\hat{I}_\infty^{f\,req}(\theta)$ is the estimation PSC obtained by the analytical formula (Eqs 7–9) with the parameter set of $\theta = \{f, U, F, D\}$ Using the trust-region algorithm, we obtain $\theta$ that minimizes Eq (10).

To assess the performance of the estimated parameters, we generated a set of synthetic data comprising five different types of STPs, ranging from strong depression to strong facilitation dynamics. These parameters were chosen from [7, 10]. Fig 2A shows the distribution of the estimated parameters using the steady-state-based method, running 100 times with different initial values. Note that the low variability in the distribution of the estimated parameters represents the accuracy of the algorithm [15, 18]. The variance of estimated parameters $F$, and $f$ were relatively higher than those of $U$ and $D$. This problem in estimating $F$, and $f$ were already reported in [12, 15]. In Fig 2B, we show that the PSCs calculated by estimated parameters (dashed lines) are closely matching with those generated by the TM model with true parameters (circles) across different stimulation frequencies.

## 2.4. Dual optimization

### 2.4.1. Challenge: Parameter estimation based on steady-state responses of postsynaptic currents does not guarantee a global solution for the TM model.

Although the steady-state-based method can accurately fit the $I_\infty$ in Eq 9 to the observed steady-state PSCs, the TM parameters acquired from this method did not always replicate the underlying transient response. In other words, the steady-state-based method might suffer from local minima. To better clarify this point, we applied the steady-state-based method to estimate the model parameters given steady-state PSCs at 8 different frequencies including 5, 10, 20, 30, 50, 100, 130, 200 Hz. With five random initializations, we achieved five different parameter sets that generate similar steady-state PSCs. However, the transient responses of the TM model using these parameters were different from each other and from those generated by original (true) parameters. Fig 3 shows an example in which the steady-state values of the analytically calculated PSCs (for different stimulation frequencies) fit those calculated by the TM model, but the transient responses are different.

Using the discrete-time TM model, one can only plot the peak of PSCs versus presynaptic spike times. As shown in Fig 3, this representation enables us to normalize the timescale of the PSC response based on stimulation pulse number, which can better illustrate both steady-state and transient responses of the PSCs for different stimulation frequencies.

### 2.4.2. Proposed solution: Fitting transient and steady-state responses of postsynaptic currents.

As observed in Fig 3, the transient and steady-state values of PSCs capture different features of the STP dynamics. Thus, applying either of them alone cannot guarantee accurate estimation. To benefit from both transient and steady-state values of PSCs, we combined the fitting algorithm for the state-state values of PSCs with a LMSE-based optimization algorithm applied to transient response of PSCs. Our proposed algorithm uses the estimated parameters calculated from steady-state values of the PSCs (see Section 2.2) in response to different frequencies of stimulation as initial parameters for a non-derivative optimization algorithm, namely, *fminsearch* (see Methods, Section 3.3.), that estimates the TM model parameters from the transient responses of the PSCs. By utilizing the new estimates as initial parameters of the trust-region algorithm for the steady-state values of PSCs, this strategy can continue to sequentially improve the accuracy of the estimated TM model parameters. We use the LMSE method to estimate the TM model parameters from transient responses of PSCs. The objective

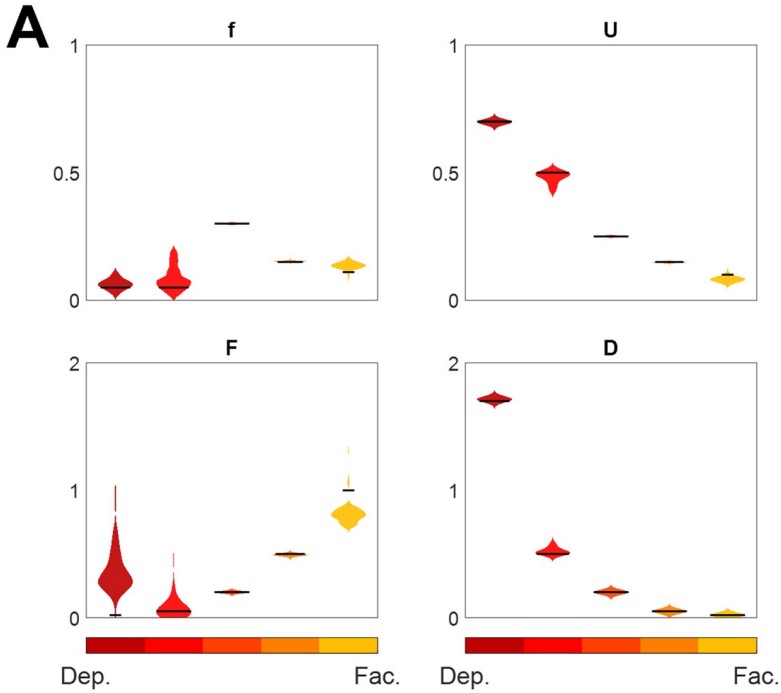

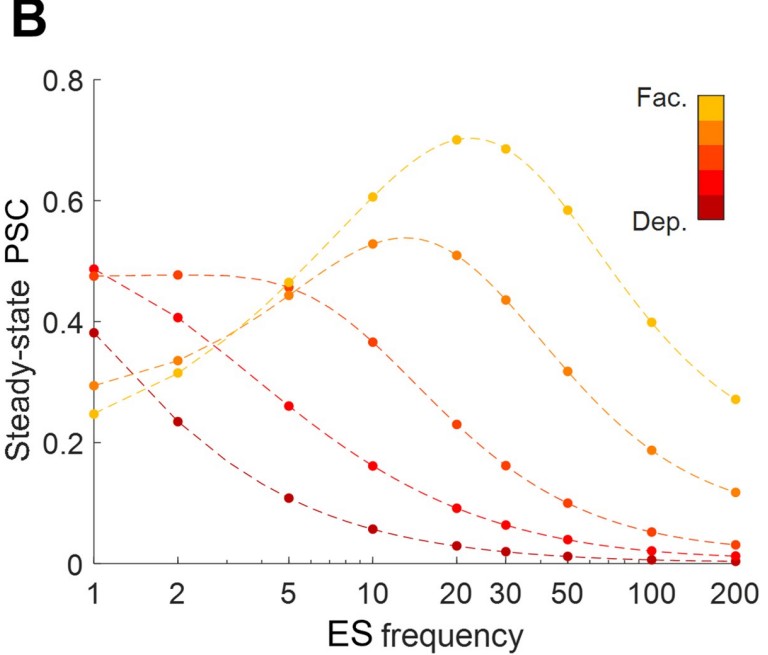

**Fig 2. Parameter estimation of TM model using steady-state responses of postsynaptic currents.** (**A**) Distribution of estimated parameters of the TM model for 5 different types of synaptic plasticity [15]. The violin plots and black bars show the distribution of estimated parameters and their true values, respectively. (**B**) Steady-state PSC in response to stimulation of different frequencies for true (dots) and a randomly selected set of the estimated parameters (dashed lines).

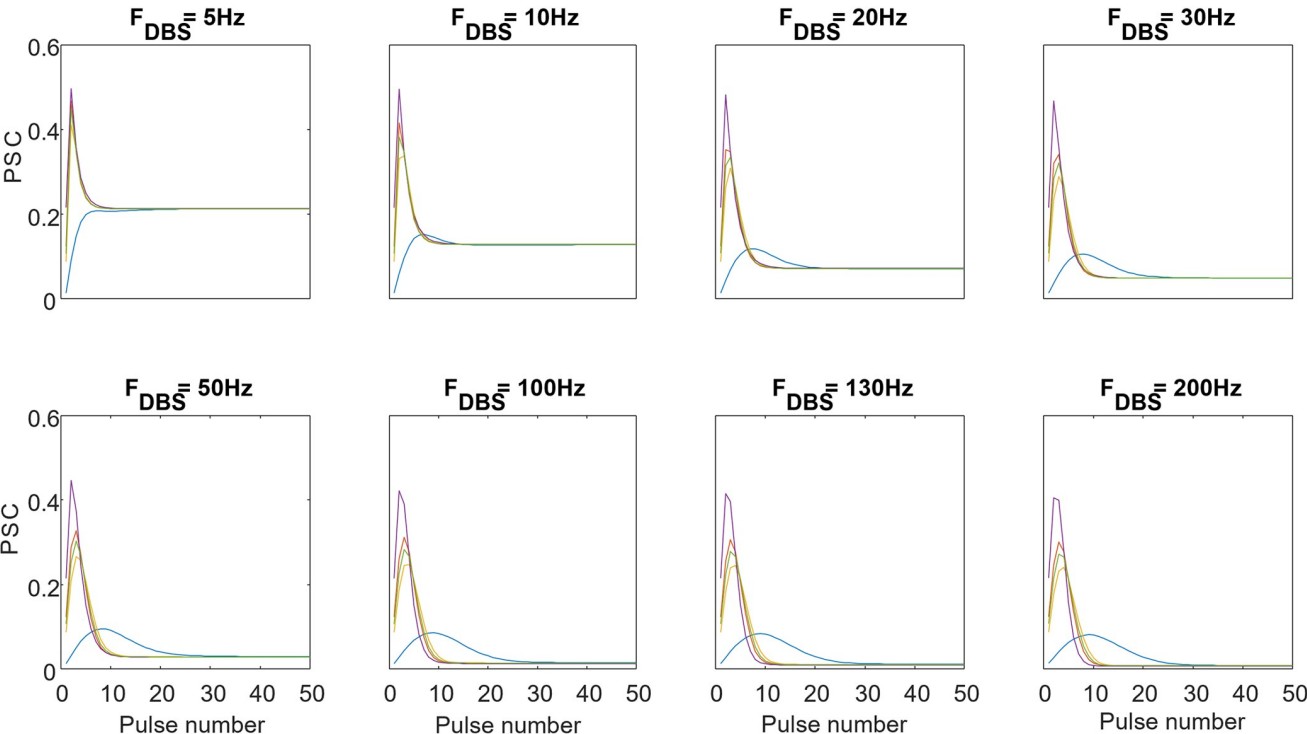

**Fig 3. Mismatch of the transient PSCs in spite of similar steady-state responses.** Each color indicates a specific type of STP and each panel represents PSC response to an individual stimulation frequency. Although the steady-state PSCs of the STP types are similar at all illustrated stimulation frequencies, the transient phases are remarkably different due to variations in the parameters sets.

function can be written as

$$\sum_{freq \in F_{ES}} \frac{1}{N_{trans}} \sum_{n=1}^{N_{trans} \triangleq 20} \left( I_n^{freq} - \hat{I}_n^{freq}(\theta) \right)^2 \tag{11}$$

where $I_n^{f\,req}$ is the peak value of PSC in response to the $n$-th stimulation pulse. $\hat{I}_n^{f\,req}(\theta)$ is the estimated PSC in the discrete-time TM model with parameter set $\theta = \{f, U, F, D\}$. The sum over the stimulation pulses was limited to $N_{trans}$, which indicates the length of the transient response of the PSC. This number can be defined mathematically as explained in Methods Section 3.2. Using various types of plasticity for different stimulation frequencies in our simulations we found that the length of the transient section is between 15 to 20 stimulation pulses, regardless of the stimulation frequency. Therefore, we used a constant number of 20 stimulation pulses as the length of the transient state in our optimization algorithm. This modification improves the speed of the LMSE method used for the transient part of the PSC and reduces convergence time in the dual optimization algorithm.

It is to be noted that conventional LMSE methods use the total length of the PSC response to estimate the TM model parameters, which can be very time consuming and computationally expensive. Moreover, the LMSE method might only fit to the steady-state values and provides a suboptimal estimation because the number of data points for steady-state responses is usually larger than that for transient ones.

The most remarkable advantage of the dual optimization algorithm to the conventional LMSE method is that the steady-state-based part of the dual optimization algorithm provides a

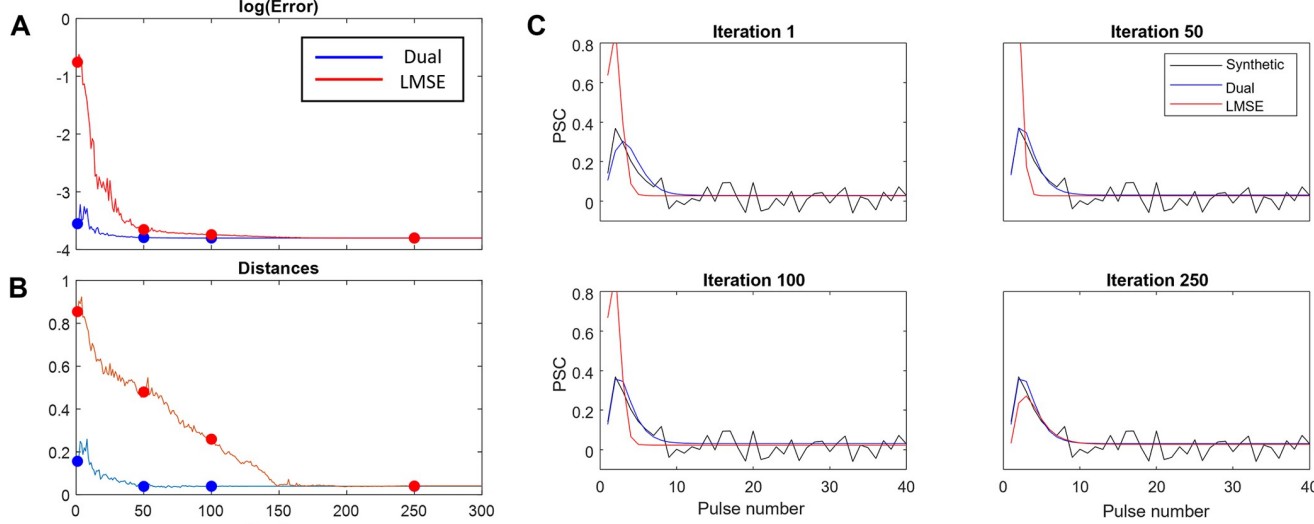

**Fig 4. Comparison between accuracy and speed of convergence for the dual optimization approach and the conventional LMSE approach. (A)** The log MSE of the model output at each iteration of the non-derivative optimization **(B)** The average distance (norm 2 of [*f*, *U*, *F*, *D*] vector) between the estimated parameter set at each iteration and the true parameter set **(C)** The peak PSC series generated by the estimated parameters at some iterations of the model.

strong initialization for the transient part. To highlight the impact of such initialization, we compared the performance of the conventional LMSE method (applied to the full length of PSC) with a random initial guess compared to the LMSE method (applied to the transient part of PSC) initialized by a set of parameters obtained from the steady-state-based method. We added a white noise with a standard deviation of 20 percent of the maximum PSC amplitude to generate noisy PSCs to assess the robustness of estimation methods in the presence of noise.

The logarithm of the MSE between the TM model output and the original PSC was calculated for 300 iterations of the *fminsearch* algorithm. Similar to the MSE, one can measure the accuracy of the estimated parameters by calculating the L2 norm between these parameters and their original values. Fig 4A and 4B respectively show that the MSE and the distance measures calculated by the dual optimization algorithms are significantly lower than those obtained by conventional LMSE methods. Moreover, strong initialization in the dual optimization algorithm results in a faster convergence compared to the conventional LMSE method. Four samples of the estimated parameters, by each method, were chosen from different iterations of the *fminsearch* algorithm to create PSC response. These responses are plotted against the original PSC in Fig 4C.

To verify the generalizability of the dual optimization algorithm for different types of STP, we generated a large synthetic dataset for PSCs by the TM model with 1000 randomly selected parameter sets. We compared the performance of the dual optimization approach with the conventional LMSE and the steady-state-based method. Fig 5 shows abstract regression plots of the three estimation approaches for each parameter of the TM model. In regression plots, the estimated parameters (y-axis) are plotted against the original ones (x-axis).

As can be seen in Fig 5, the dual optimization approach remarkably increases the accuracy (the samples in the y-axis have the same values as those in the x-axis) and precision (i.e., standard deviation reflected by the shaded areas) of all estimated parameters compared to the other algorithms. Specifically, the parameter *F* was estimated more accurately by the dual optimization algorithm compared to that by the LMSE method. To better clarify the accuracy and

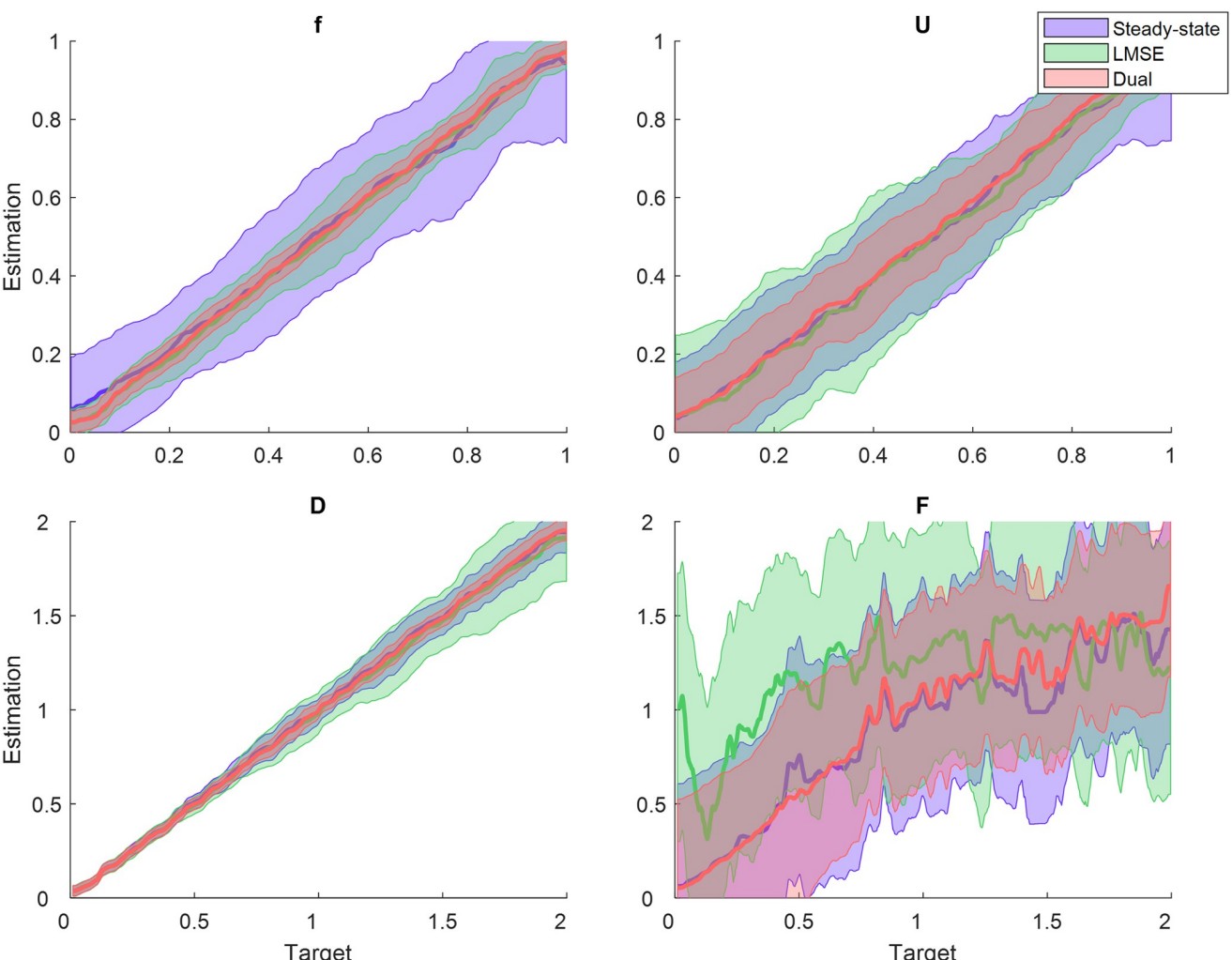

**Fig 5. Comparison between the performance of the steady-state-based, the conventional LMSE, and the dual optimization approaches in estimating parameters of the TM model.** Green, blue, and red colors represent the steady-state-based, the conventional LMSE, and dual optimization approaches, respectively. The thick lines represent smoothed moving average (Hann window) of the estimation. The shaded areas show the 68.2% confidence interval of the estimated values. As the estimation regression line gets closer to the $y = x$ line the correlation of the estimation and true values increases, and the estimation is more accurate.

precision of the estimated parameters, we chose one sample of the synthetic dataset together with corresponding estimated parameters and showed their distributions. In Fig 6A, the mean and the standard deviation of the estimated parameters indicate the accuracy and precision of the algorithm, respectively. As it is obvious in this figure, the dual optimization algorithm out-performs the conventional LMSE and steady-state based methods. Neither MSE-based nor steady-state-based approaches achieve the optimum solution, and they suffer from local optima. An example of PSCs generated by the estimated parameters obtained from each algo-rithm was shown against the synthetic PSCs in Fig 6B at three stimulation frequencies. The above three algorithms successfully predicted the steady-state values of the PSC, despite the variability of their estimated parameters. However, as expected, the steady-state-based method failed to fully predict the transient PSCs. Although the estimated parameters obtained by the dual optimization method generated fairly close outputs to that of the LMSE method, the

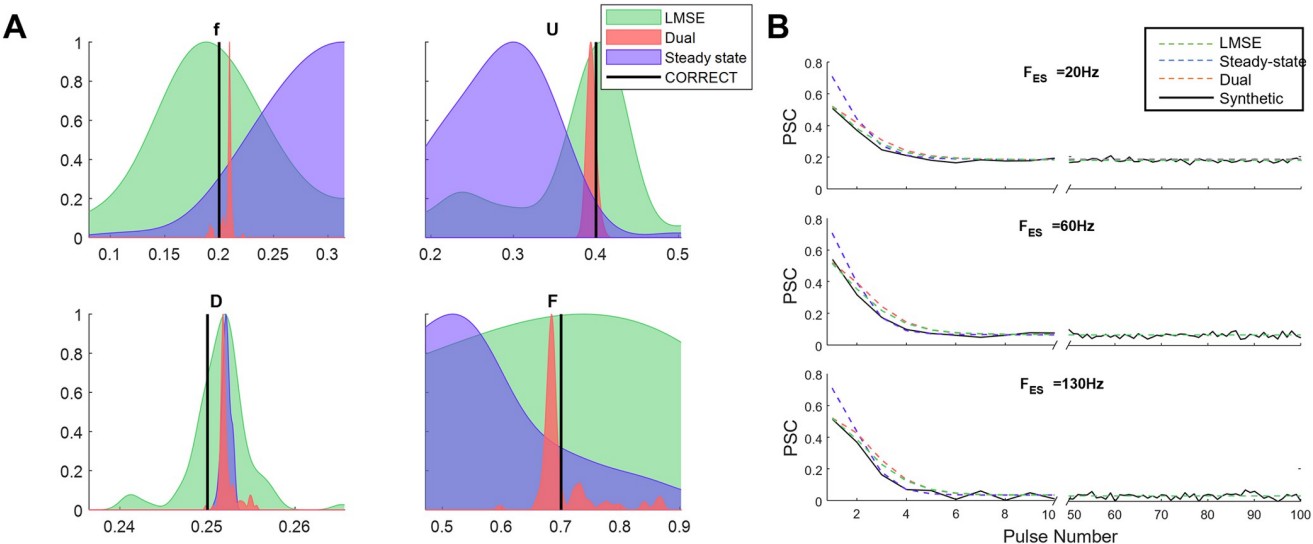

**Fig 6. Distribution of estimated parameters by the three approaches. (A)** Normalized probability distribution function (PDF) of the estimated parameters by the three approaches. The PDFs have been estimated from the results of applying each algorithm with 1000 different initializations. Moreover, a 5% white noise has been added to the reference data. The dual optimization approach improved the estimation precision by decreasing the standard deviation of the PDF and moving the mean of the PDFs closer to the true values. Note that the PFDs of the dual optimization approach are extremely narrow that might be covered by the true value lines. The x-axis range is also subjective to only provide a clear illustration thus the distance between the peak of PDFs in the upper right panel (D), is lower than it appears in the figure. **(B)** The discrete-time TM model output generated by the estimated parameters that are obtained from the three methods. The plots include the trace of PSC response for both transient state (from stimulus 0 to 10) and steady-state (last 50 stimuli).

estimated parameters of dual optimization method are significantly more precise and closer to the correct parameters underly the synthetic data.

## 2.5. An alternative solution in case of lack of information

Although the dual optimization approach has several advantages, the number of data points required for estimating the parameters must be at least as many as the number of the model parameters (in the case of four-parameter TM model we require 4 recorded DBS frequencies). However, the available experimental data (see Section 2.6) only provides the PSC recordings during 3 DBS frequencies, which can be compensated in fitting the transient part of the PSC response. To estimate the TM parameters from experimental recordings from 3 frequencies, we modified the dual optimization by freezing one of the parameters during the fitting of the steady-state PSC. Since the parameter $U$ has the least effect on changing the steady-state PSC, we choose it as the frozen parameter in the algorithm. This modified version of the dual optimization approach has an imbalance pace in fitting the steady-state versus the transient part of the response. Therefore, we should decrease the number of optimization steps in both steady-state and transient optimizations and increase the iterations between these two steps until convergence. This modification prevents the optimizer from overfitting and becoming trapped in local minima. Fig 7 shows the distribution of the estimated parameters with 1000 different initial guesses. Since in optimizing the steady-state the parameter $U$ is frozen, the algorithm cannot estimate the correct value of the model parameter. However, dual optimization provides a reliable result compared with the other methods.

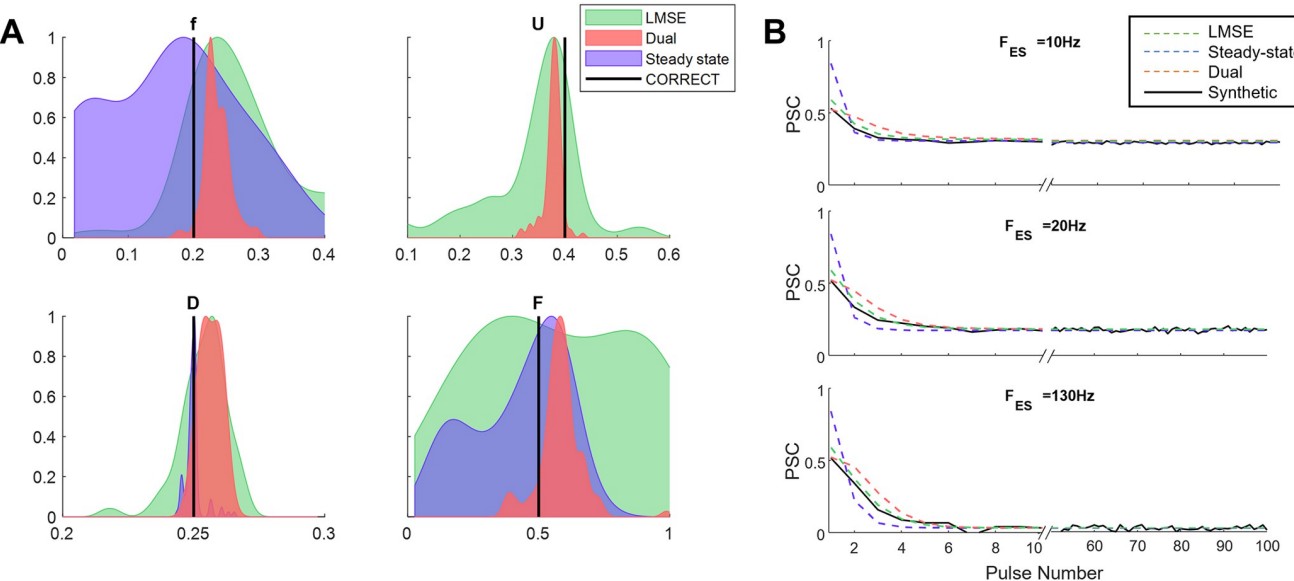

**Fig 7. Distribution of estimated parameters by the three approaches with only 3 recorded responses. (A)** Normalized PDF of the estimated parameters by the three approaches. The PDFs have been estimated from the results of applying each algorithm with 1000 different initial guesses. Moreover, a 5% white noise has been added to the reference data. The dual optimization approach improved the estimation precision by decreasing the standard deviation of the PDF and moving the mean of the PDFs closer to the true values. **(B)** The discrete-time TM model output generated by the estimated parameters (in case of having the only the observed frequency) that are obtained from the three methods. The plots include the trace of PSC response for both transient state (from stimulus 0 to 10) and steady-state (last 50 stimuli).

## 2.6. Estimating the TM parameters from in vitro recording of rat STN neurons

To test the performance of the algorithm in experimental data, we applied the dual optimization approach to in vitro recordings of STN from a single juvenile transgenic Wistar rat. In this experiment (see Methods, Section 3.6.), efferent axons that are connected to STN neurons were activated by extracellular stimulation pulses. The details of the experimental procedure are mentioned in [7], and the Methods section.

To apply the dual optimization method on the experimental data, it is necessary to adjust the hyper-parameters of the algorithm including the number of iterations over transient and steady states, maximum number of steps for steady-state optimization, maximum steps for transient optimization, and the factor of penalizing distance from steady-state estimation. These hyper-parameters were obtained by trial and error to acquire the best results. It is to be noted that the dual optimization algorithm was robust for a wide range of hyper-parameters and generated reasonably accurate PSCs.

As shown in Fig 8A, The PSCs generated by the estimated parameters precisely matched to the PSCs recorded experimentally for three different frequencies of the electrical stimulation. It is worth highlighting that estimated parameters were consistent across frequencies, confirming that the TM model with accurate parameters captures the dynamics of STP. Fig 8B shows the distribution of the TM model parameters by running the algorithm from 100 different initial guesses. The low variance of the distributions shows that the results are repeatable and the estimation is precise. Note that we used a logarithmic scale to show the distribution of parameters because the standard deviation of the parameter *f* was extremely low, and a logarithmic scale provides a better resolution for lower values.

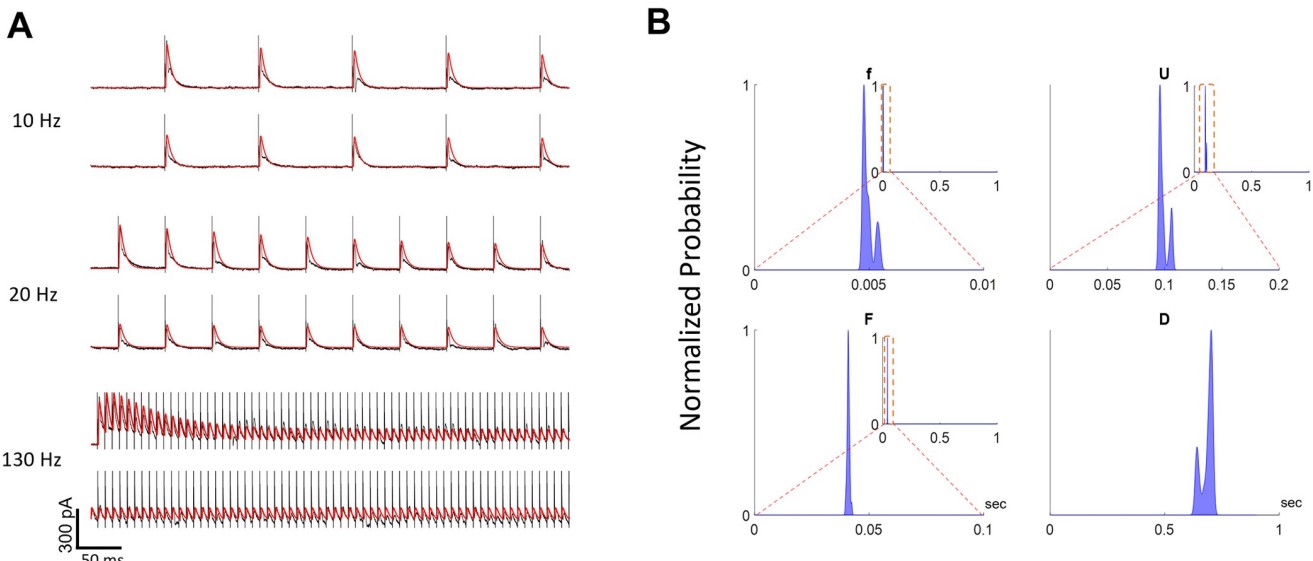

**Fig 8. Inferring the model parameters from the experimental data.** (A) The output generated by the estimated parameters perfectly fitted the experimental data. (B) Distribution of the estimated parameters by 100 trials of the dual optimization approach with different initial guesses. The estimated parameters are low variance which shows the confidence of the method.

**2.6.1. Measurement noise in the experimental data.** Measurement noise is an inevitable source of uncertainty in neural recordings that might lead to inaccurate parameter estimation. Various methods were developed to reduce the impact of such noise (including power line noise) in signals (e.g., see [24, 25]). In the context of inferring parameters of a dynamical system from noisy observation, it was recently shown that using approaches based on numerical integration (instead of the differentiation) can improve parameter estimation task [26, 27]. Although the recorded PSC might be contaminated by measurement noise, one advantage of the steady-state estimate, in the dual optimization approach, is that the steady-state PSC in response to DBS pulses can be inferred from the average (over time) of recorded PSC, making our approach reasonably robust to the measurement noise.

## 3. Methods

### 3.1. Discrete-time Tsodyks-Markram model of STP

Despite intrinsic spike firing of the presynaptic neuron in stimulation-induced STP, it is assumed that all evoked spikes are equal during each stimulation pulse. Since the presynaptic firing events happen every $\frac{1}{F_{DBS}}$ $sec$, we can only calculate the state of the variables ($u$, $r$, $I$) at $t_{spk} \in \left\{ \frac{1}{F_{DBS}}, \frac{2}{F_{DBS}}, \frac{3}{F_{DBS}}, \ldots, \frac{n}{F_{DBS}} \right\}$. Hence, the discrete-time TM model or $I[n] = I\left(\frac{n}{F_{DBS}}\right)$ can be described by a recursive function. We started by defining the following equations:

$$u(t) = \Delta u_n \sigma(t_n) + u_n(t), \tag{12}$$

$$\Delta u_n \triangleq u(t_n^+) - u(t_n^-), \tag{13}$$

$$u_n(t) \triangleq u(t) \quad ; \quad t_{n-1} < t \leq t_n, \tag{14}$$

where $u(t)$ in Eq 12 is defined as summation of the rapid changes $\Delta u_n$ at the stimulation arrival time $t_n$ (Eq 13), and the exponential dynamics between the stimulation pulses were denoted as

$u_n(t)$ (Eq 14). We calculated both parts as follow to infer Eq 15 for the model states at the time $t_n^+$ which is right after arrival of the $n$-th stimulation pulse:

$$
\begin{aligned}
\Delta u_n \ & = u(t_n^+) - u(t_n^-) \\
& = \lim_{\sigma \to 0} \int_{t_n-\sigma}^{t_n+\sigma} \left( \frac{U - u(t)}{F} + f \times (1 - u(t_n^-)) \right) \times \delta(t - t_{sp})) dt \\
& = f \times (1 - u(t_n^-)) \\
& \qquad u(t_n^+) = f + (1-f) \times u(t_n^-)
\end{aligned}
\tag{15}
$$

As the first term inside the integral (Eq 15) has no Dirac delta component, it can be ignored in calculating the integral in the narrow boundaries of $t_n-\sigma$ to $t_n + \sigma$. A similar approach can be applied for the state value between to stimulation pulses to infer the Eq 16 which shows formulate $u(t_n^-)$ based on the $u(t_{n-1}^+)$:

$$
\begin{aligned}
u_n(t) \ & = \int_{t_{n-1}^+}^{t_n^-} \left( \frac{U - u(t)}{F} + f \times (1 - u(t_n^-)) \times \delta(t - t_{sp}) \right) dt \\
& = \int_{t_{n-1}}^{t_n} \frac{U - u(t)}{F} dt \\
& = U + (u(t_{n-1}^+) - U) exp\left( \frac{-(t - t_{n-1})}{F} \right), \\
u_n(t_n^-) \ & = u_n(t = t_n^-) \\
& = U + (u(t_{n-1}^+) - U) exp\left( \frac{-(t_n - t_{n-1})}{F} \right), \\
u_n(t_n^-) & = U + (u(t_{n-1}^+) - U) exp\left( \frac{-1}{F_{DBS} \times F} \right).
\end{aligned}
\tag{16}
$$

Similar to Eq 15, the Dirac delta function in the second component of the integral in Eq 16 has zero value in the boundary of $t_{n-1}^+$ to $t_n^-$ thus that term can be removed. By combining Eqs 15 and 16, we achieve the recursive relation between $u(t_n^+)$ and $u(t_{n-1}^+)$ which can be used as the definition of the discrete-time TM model (Eq 18):

$$
u(t_n^+) = f + (1-f)u(t_n^-),
$$

$$
u(t_n^+) = f + (1-f)\left( U + (u(t_{n-1}^+) - U) exp\left( \frac{-1}{F_{DBS}\tau_{Fac}} \right) \right).
\tag{17}
$$

$$
u[n] \triangleq u(t_n^+),
$$

$$
u[n] = f + (1-f)\left( U + (u[n-1] - U) exp\left( \frac{-1}{F_{DBS}\tau_{Fac}} \right) \right).
\tag{18}
$$

A similar process can be done on the state variable $r(t)$. Similarly, we started by defining the following equations:

$$
r(t) = \Delta r_n + r_n(t),
\tag{19}
$$

$$
\Delta r_n \triangleq r(t_n^+) - r(t_n^-),
\tag{20}
$$

$$
r_n(t) \triangleq r(t) \quad ; \quad t_{n-1} < t \le t_n,
\tag{21}
$$

where $r(t)$ in Eq 19 is the summation of the rapid changes $\Delta r_n$ at the stimulation arrival time $t_n$ (Eq 20), and the exponential dynamics between two stimulation pulses were denoted as $r_n(t)$ (Eq 21). We calculated both parts as follow to infer Eq 22 for the model states at the time $t_n^+$ which is right after arrival of the $n$-th stimulation pulse:

$$
\begin{aligned}
\Delta r_n &= r(t_n^+) - r(t_n^-) \\
&= \lim_{\sigma \to 0} \int_{t_n - \sigma}^{t_n + \sigma} \left( \frac{1 - r(t)}{D} - u(t^-)r(t^-)\delta(t - t_{spk}) \right) dt,
\end{aligned}
$$

$$
r(t_n^+) = r(t_n^-)\left(1 - u(t_n^-)\right), \tag{22}
$$

A similar approach can be applied for the state value between two stimulation pulses to calculate $r(t_n^-)$ based on the $r(t_{n-1}^+)$. By replacing $r(t_n^-)$ and $u(t_n^-)$ in the Eq 22, discrete representation of the state $r$ can be calculated as follow. Note that similar to Eq 16, the second term inside the integral in Eq 22 is removed because the Dirac delta function has zero value within the integral boundaries:

$$
\begin{aligned}
r_n(t) &= \int_{t_{n-1}^+}^{t_n^-} \left( \frac{1 - r(t)}{D} - u(t^-)r(t^-)\delta(t - t_{spk}) \right) dt \\
&= \int_{t_{n-1}}^{t_n} \frac{1 - r(t)}{D} dt \\
&= 1 - (1 - r(t_{n-1}^+))exp\left( \frac{-(t - t_{n-1})}{D} \right),
\end{aligned}
$$

$$
r(t_n^-) = 1 - (1 - r(t_{n-1}^+))\exp\left( \frac{-(t_n - t_{n-1})}{D} \right),
$$

$$
r(t_n^-) = 1 - (1 - r(t_{n-1}^+))\exp\left( \frac{-1}{F_{DBS} \times D} \right), \tag{23}
$$

$$
r[n] \triangleq r(t_n^-),
$$

$$
r[n] = 1 - (1 - r[n-1] \times (1 - u[n-1]))\exp\left( \frac{-1}{F_{DBS} \times D} \right). \tag{24}
$$

The discrete-time TM model will be completed by:

$$
I[n] = I[n-1]\exp\left( \frac{-1}{F_{DBS} \times \tau_{syn}} \right) + Au[n]r[n]. \tag{25}
$$

## 3.2. Calculating the steady-state value

By stimulating the presynaptic neuron with homogeneous pulses and having homogenous action potential at the synaptic terminal the postsynaptic currents in response to these action potentials settle down to a steady-state value. This value is a function of TM model parameters and the stimulation frequency. We defined the steady-state values of $r$, $u$, and $I$ in the discrete

TM model as follow:

$$u[n] = u[n+1] = u_\infty, \tag{26}$$

$$R[n] = R[n+1] = R_\infty, \tag{27}$$

$$I[n] = I[n+1] = I_\infty \tag{28}$$

where $t_n$ represents the time of the $n$-th spike and $u_\infty$, $r_\infty$, and $I_\infty$ are the steady-states of the discrete-time TM model. The steady-state value of postsynaptic current can be calculated:

$$u_\infty = \frac{f + (1-f) \times U \times \left(1 - exp\left(\frac{-1}{F_{DBS} \times F}\right)\right)}{1 - (1-f) \times exp\left(\frac{-1}{F_{DBS} * F}\right)}, \tag{29}$$

$$r_\infty = \left(\frac{1 - exp\left(\frac{-1}{F_{DBS} \times D}\right)}{1 - (1 - u_\infty) \times exp\left(\frac{-1}{F_{DBS} \times D}\right)}\right) \tag{30}$$

$$I_\infty = \frac{A \times u_\infty \times r_\infty}{1 - exp\left(\frac{-1}{F_{DBS} \times \tau_{syn}}\right)} \tag{31}$$

## 3.3. Optimization and cost function

To fit the model output to the experimental data or synthetic reference data, the parameter space should be explored to find a set of parameters that generates the most similar output to the reference data. Assuming that the noise/measurement error has a Gaussian distribution, we used MSE to calculate dissimilarity between the model output and the reference. Since we inferred the analytical formula for the steady-state response PSC, calculating its gradient is feasible, therefore we can apply gradient-based optimization algorithms. However, since the transient response of the PSC is provided by simulating the TM model, calculating the derivative of the MSE over transient PSC is computationally intensive and impractical. Therefore, we used a non-derivative optimization algorithm for minimizing this error.

To minimize the error of the steady-state estimation, we applied the Trust-Region optimization methods using built-in functions of MATLAB software. This method uses first- and second-order derivatives of the cost function to calculate Tylor estimation of the values of the loss function in a neighborhood points of the current parameters. It then updates the parameters to the point with the least value and continues until the convergence or the maximum number of iterations reaches. The trust-region algorithm is faster than non-derivative search-based algorithms, because non-derivative algorithms call the loss function and wait for its results for every single point. While the trust-region algorithm only calculates the first and second derivatives and estimates all the other points using matrix multiplications all at once [28]. This makes the trust-region algorithm faster than non-derivative search-based algorithms.

Although derivative-based algorithms are generally faster than non-derivative search-based algorithms, they require the mathematical formula of the loss function to calculate the derivatives. We successfully calculated the mathematical formula for the steady-state values of postsynaptic currents in the TM model. However, the equations for the transient state are more complex and we cannot employ the trust-region method or any other derivative based

algorithms for fitting this part. Instead, we used Nelder-Mead optimization to optimize the transient part of the PSC response. The Nelder-Mead method starts from a set of random possible answer and evaluates them by calling the loss function. The algorithm starts with a set of possible answers and optimizes the set by replacing the worst member with a better value. The substitute point is determined by moving the worst point toward the average point of all other members of the set. This updating continues until reaching the convergence or the maximum number of iterations. This algorithm requires function evaluation at each step and typically it is not as fast as derivative-based algorithms [29, 30].

### 3.4. Detecting the transient PSCs

As we described in the result section, transient and steady-state parts of the PSC response have different sensitivity to TM parameters, which introduce an error on the whole time trace of the response and decrease the estimation accuracy of some parameters. Therefore, to fit both of these parts correctly in the dual optimization algorithm, we calculated the transient part of postsynaptic response for each individual frequency of electrical stimulation as follow:

$$Transient\ PSCs = \{I[1],\ I[2],\ \ldots,\ I[N_{trans}] \mid \forall n \geq N_{trans}\ ;\ I[n] - I[n-1] < 0.05 \times I[n-1]\} \quad (32)$$

where $N_{trans}$ is referred to the stimulation number corresponding to the end of the transient part of the PSC. $N_{trans}$ is defined as the least $n$ that for every $n$ after it the PSP has less than %5 changes. The criterion defined in Eq 32 must be applied to the reference data and can be adjusted as a hyper-parameter of the algorithm. In both synthetic and experimental data, we observed that the steady-state part begins from the $N_{trans} = 15$ for most of the stimulation's frequencies. Thus, $N_{trans} = 15$ can be considered as a rule of thumb for distinguishing between transient and steady-state parts of the PSC.

### 3.5. Iterative dual optimization algorithm

Since the sensitivity of each part of the response to the model parameters is different, overfitting the model to any of these parameters interferes with the estimation of parameters with lower sensitivity. Therefore, we have to constrain both optimizers (for transient and steady-state parts of the PSC) to avoid overfitting. To control the optimizers, we constrain the number of iterations inside each of them and repeat the algorithm of fitting the steady-state and transient state multiple times. Moreover, we penalize the distance from the initial guesses in fitting the transient part to eliminate big jumps from the values estimated by the steady-state-based part.

### 3.6. Methods for experimental data

Methods for collection of experimental data shown in Fig 8 have been previously described in detail in [7]. 300-μm-thick brain slices containing the STN were cut from acutely isolated rat brains. The slice is provided from a juvenile (P14-P21) transgenic Wistar rat. Whole-cell patch-clamp recordings were performed in a submerged-type recording chamber continuously perfused with artificial cerebrospinal fluid (containing the following (in mM): 126 NaCl, 2.5 KCl, 1.2 NaH2PO4, 11 glucose, 19 NaHCO3, 2.4 CaCl2, 1.2 MgCl2) held at 34˚C. Somatic whole-cell patch-clamp recordings were performed using pipettes pulled from borosilicate glass capillaries (2 mm outer/1 mm inner diameter) on a horizontal puller (P-97, Sutter Instrument). The pipettes were filled with an intracellular solution containing the following (in mM): 145 K-gluconate, 6 KCl, 10 HEPES, 0.2 EGTA, 5 Na2-phosphocreatine, 2 Na2ATP, 0.5 Na2GTP, and 2 MgCl2 (290–300 mOsm, pH adjusted to 7.2 with KOH). Filled pipettes had a

resistance of 3–7 MΩ. For extracellular stimulation, a tungsten bipolar electrode (tip diameter 30 μm) was placed in the rostral part of the STN. The electrode was lowered 50 μm into the slice. For the experiment shown stimulation intensity was set to and 500 μA, with a pulse duration of 100 μs. To study frequency-dependent dynamics of synaptic inputs to STN neurons, stimuli were applied at 10, 20, and 130 Hz. Each stimulation train was applied for 1 s, and the stimulation interval was followed by a 4 s break. Thus, the total sweep duration was 5 s. A total of 10 sweeps were recorded for each stimulation frequency. The four neurons displayed were recorded simultaneously in a single experiment.

### 3.7. Offline analyses

To fully replicate the PSC in the experimental study, we convolved a double exponential kernel to match with the empirical recordings. Fig 9 demonstrates the double exponential kernel that is used for replicating the PSC compared to the experimental recordings. Note that the recordings consist of a compound effect of the excitatory and inhibitory trans-synaptic inputs. This could potentially challenge the design of the estimation algorithm because excitatory and inhibitory inputs have different STP dynamics, and their compound effect cannot necessarily be separated. As discussed in [7], the rat STN consists of multiple regions that have different proportions of excitatory and inhibitory inputs. Therefore, to minimize the problem of compound inputs, we chose the recordings with dominant inhibitory inputs. Furthermore, Steiner et al. [7] proved experimentally using neuroreceptor blockers, that the excitatory and inhibitory responses appear in the recorded PSC response, as separatable negative and positive peaks, respectively. Since the kernel of excitatory PSCs are different from that of the inhibitory, we can easily distinguish between these two responses by a visual analysis of the recorded

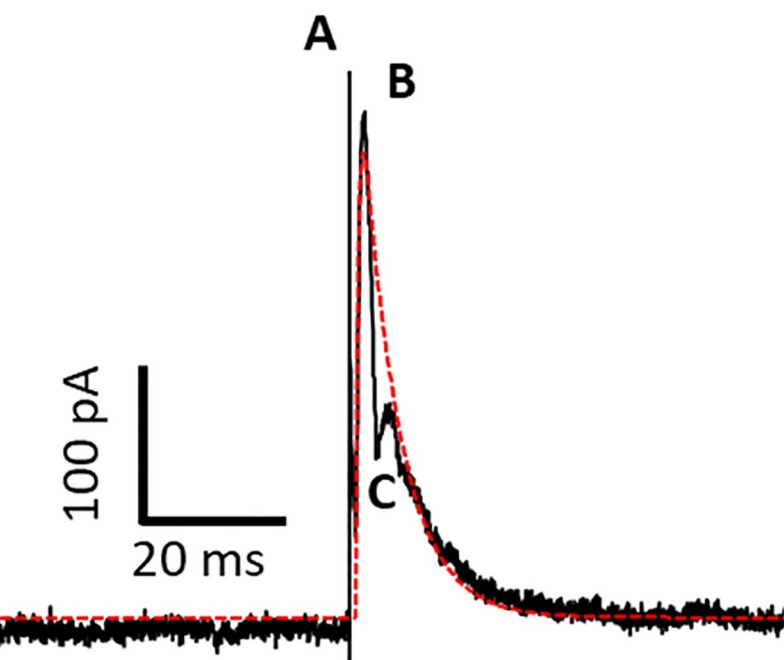

**Fig 9. Generating the PSC kernel using a double exponential function.** The red line shows the estimated kernel generated by a double exponential function. Black line represents the experimental recordings of the acute slice of the rat STN. **A** is the stimulation artifact; **B** is the positive peak of representing the inhibitory response and **C** is the negative (reversed) peak representing the excitatory response. The difference between PSC signal and the estimated PSC kernel, which is maximized at the reversed excitatory peak, can be considered as the effect of the excitatory inputs.

signal. The estimated PSC kernel and the recorded signal of the PSC in response to a single DBS pulse is shown in Fig 9. The chosen trial has predominant GABAergic inhibitory inputs, which appeared as the positive peak in the signal. The effect of the excitatory input is slower than that of inhibitory, which is observable as a reversed peak in the PSC current.

By separating the inhibitory response from the excitatory, we can model the STP dynamics with the four-parameter Tsodyks-Markram model. Although there is more than one excitatory synapse responsible for the generation of the PSC, it is assumed that all excitatory synapses follow the similar STP dynamics. In [2], the presynaptic inputs were simulated by multiple Tsodyks-Markram models and the effect of axonal failure could be adjusted in the model [2, 11]. In this method, the effect of axonal failure can be easily incorporated by the absolute synaptic activation parameter ($A$ in Eq 3).

The code for the dual optimization algorithm is available at: https://github.com/nsbspl/Daul_Optimization.

## 4. Discussion

STP induced by DBS-like stimulation was observed in both human and animal studies [2, 7]. By utilizing the well-known Tsodyks and Markram (TM) model of STP, we developed a novel algorithm to infer model parameters from the postsynaptic current (PSC) of a stimulated neuron in response to DBS-like stimulations with different constant frequencies. We used conceptual findings from a recent theoretical work [2] on the cellular mechanism of DBS to propose a parameter estimation algorithm that incorporates stimulation pulses as pre-synaptic spikes. Unlike other model-based estimation methods for STP, we distinguished between the steady-state and transient responses of recorded PSCs and estimated TM model parameters in two steps. First, the steady-state PSC was calculated for the extended TM model (four-parameter TM model) analytically. We fit the analytical steady-state PSC to that obtained from recorded PSCs from stimulations with different frequencies in the interval of [0–100] Hz. Second, we optimized TM model parameters by fitting the transient part of the model to that obtained by PSCs across all stimulation frequencies utilizing parameters estimated in the first step as initial values of a non-derivative optimization algorithm. This two-step algorithm was referred to as dual optimization algorithm.

Using extensive synthetic data, we demonstrated that the performance of dual optimization method was significantly better than that of conventional LMSE method. Specifically, we showed that the dual optimization algorithm reliably estimated TM parameters in the presence of noise. To further validate the performance of dual optimization algorithm for experimental data, we applied our algorithm to PSCs recorded from acute rodent brain STN slices during DBS-like stimulation with three different frequencies [7]. We showed that reconstructed PSCs with estimated parameters were accurately fit to recorded PSCs from STN neurons with inhibitory-dominant inputs. The proposed dual optimization algorithm provided a strategy to illustrate the dynamics of stimulation-induced STP with the TM model. As the TM model is a phenomenological model, the estimated stimulation-induced STP dynamics describe the interaction between activated presynaptic afferents and postsynaptic responses of the stimulated neuron. Therefore, it is essential to obtain STP parameters that are consistent across different frequencies of stimulation. To the best of our knowledge, the dual optimization algorithm is first in its kind that infers stimulation-induced STP from recorded PSCs in response to different frequencies of stimulation.

Although the idea of dual optimization has not been directly used in the previous works, Costal et al. [12] showed that increasing the length of the stimulation can increase the accuracy of the estimated parameters of the TM model. However, after 20 to 50 pulse the increase of

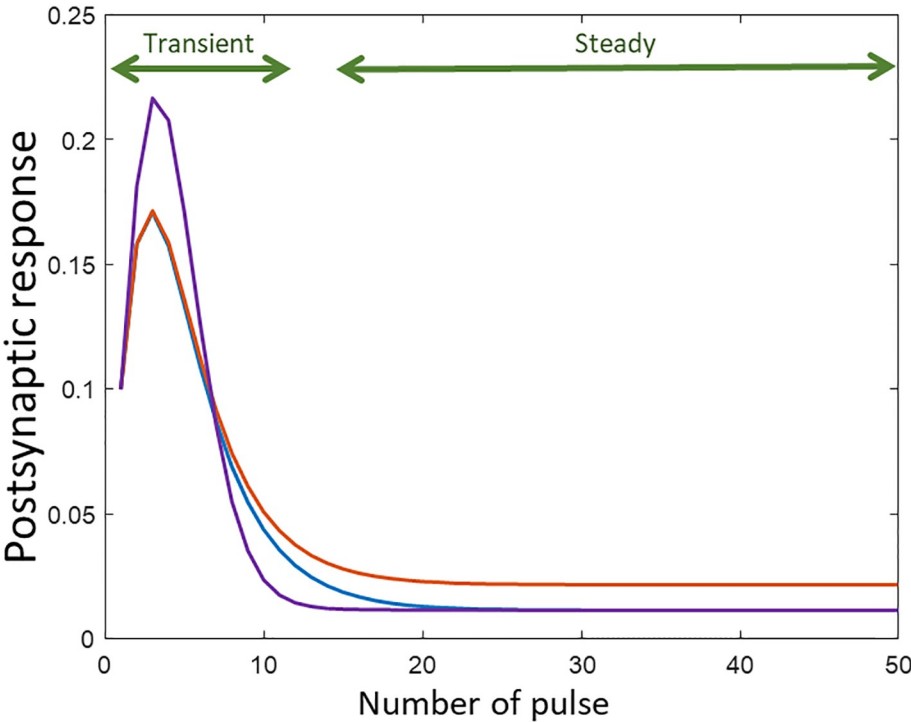

**Fig 10. The sensitivity of the transient and steady-state section of the postsynaptic response to changing the TM model parameters.** The postsynaptic response of the discrete-time TM model to 50 Hz periodic pulse stimulation. The parameters $F$, and $D$ are changed to demonstrate the sensitivity of the different sections of the PSC response to these parameters. The transient section is considered as the first part of the response that maximum EPSC is still changing for each spike. The steady section is the following part that PSCs converge to a value and remain unchanged. Considering $\theta = [F, D, f, U]$ the vectorized form of the parameters: $\theta_{blue} = [2, 0.05, 0.1, 0]$, $\theta_{red} = [1, 0.05, 0.1, 0]$ and $\theta_{purple} = [2, 0.5, 0.1, 0]$. Considering the blue trace as the reference case changing decreasing $D$ leads to increase in the steady-state value in the red trace. As it is shown despite changing $D$ the transient sections of blue and red traces are almost similar. The purple trace is generated by a larger $F$ which leads to change in the transient state of the PSC response while having a steady-state value close to that of the blue trace.

stimulation length did not improve the estimation accuracy. This observation is consistent with the fact that the PSC response to DBS-like stimulations reach a steady-state value after around 20 pulse and no more information adds to the observation. It can be shown that the transient- and steady-states of a PSC response to DBS-like stimulation have various sensitivity to the TM model parameters. As shown in Fig 10, decreasing the depression time constant, $D$ (see Eq 2), does not significantly affect the transient part of the PSC response, but it increases the steady-state value. On the other hand, changing the facilitation time constant, $F$ (see Eq 1), greatly affects the transient part of the PSC while the steady-state part remains almost unchanged.

Since the steady-state of the PSC is less sensitive to the facilitation time constant (parameter $F$), the MSE cost function defined on the whole length of the PSC is less sensitive to this parameter. Thus, for the PSC responses with long lengths (approximately with more than 30 pulses), the conventional LMSE method is not successful to infer this parameter accurately.

Furthermore, Costa et al. [12] showed that using a Poisson process as the presynaptic simulation significantly improves the accuracy of the parameter estimation. It was discussed that the Poisson train improves the accuracy of estimated parameters because, unlike the periodic train, the PSC does not reach a steady-state value. Although available DBS-like stimulations in

the present study had constant inter-pulse intervals (similar to periodic stimulations), incorporating all individual frequencies of stimulations in the dual optimization algorithm enhanced the accuracy of estimated parameters.

Despite the robustness of the dual optimization algorithm for estimating parameters of the TM model, characterizing dynamics of stimulation-induced STP may involve several challenges and limitations. First, the TM model parameters underlying stimulation-induced STP do not necessarily describe the dynamics of specific types of STP which exist in various synapses of stimulated neurons. As discussed in [2], each DBS pulse can simultaneously activate various presynaptic inputs of the stimulated nucleus, thus one cannot expect that the estimated TM model parameters during electrical stimulation lie within a range of STP values calculated for glutamatergic and GABAergic synapses in double cell patch clamping experiments [6]. Nevertheless, using a well-known model for STP to characterize changes to postsynaptic currents (or potentials) of a neurons in response to different frequencies of electrical stimulation provides a standard approach to represent dynamics of STP induced by these stimulations.

Second, electrical stimulation is not the only source of variation in the recorded postsynaptic responses, the stochasticity of vesicle release and changes in the excitability of the postsynaptic neuron might alter the statistics of the postsynaptic responses [15]. Ghanbari et al [15] studied how these sources of variability affect estimation of the STP parameters (given presynaptic and postsynaptic spikes) by introducing each of these variables separately [15]. It was shown that changes in the excitability of the postsynaptic neuron–modelled by after-hyperpolarization current to the postsynaptic neuron–does not significantly change STP estimation whereas stochastic vesicle release can lead to biases in the estimated STP parameters. To validate the accuracy of estimated STP parameters in the present study, we added white noise with a standard deviation of 20 percent of the maximum amplitude of the PSC and tested if the dual optimization algorithm reliably estimated STP parameters. As shown in Fig 4, the STP parameters estimated by the dual optimization algorithm were accurate, and significantly more reliable than those obtained by conventional LMSE algorithm in the presence of noise.

In addition to abovementioned sources of variability in the postsynaptic responses, the impact of other types of plasticity (e.g., long-term depression [31] and long-term potentiation [32]) and other postsynaptic factors like desensitization [33, 34], depolarization blockade [35], or saturation of postsynaptic receptors [36] can vary synaptic weights in time scales that are different, but not necessarily distinct, from those related to the STP. Therefore, these factors influence the interaction between the pre- and postsynaptic neurons which cannot be captured by the dynamics of STP solely. It was suggested that alternative models of plasticity with less biophysical constraints like generalized bilinear model [15] and linear-nonlinear cascade model [19] could provide more flexible representation of STP dynamics compared to the TM model. Despite such limitations, the conventional three-parameter TM model is sufficient to describe STP in many cases [12]. Here, we used a four-parameter TM model [14] that provided more degrees of flexibility compared to the conventional one.

In the context of electrical stimulation with constant frequencies, we discussed that at least 4 different individual frequencies were required to reliably estimate the TM model parameters from the steady state response of the recorded PSCs. However, the proposed dual optimization algorithm can still be useful when less than 4 individual frequencies of electrical stimulation are available. As demonstrated in Section 2.6, one can freeze the parameter $U$ in the four-parameter TM model and run the dual optimization algorithm with more iterations between the steady-state and transient parts of the postsynaptic responses to achieve sufficiently accurate estimates. Using the synthetic dataset, we demonstrated that this modification does not affect the accuracy of the algorithm.

## Author Contributions

**Conceptualization:** Alireza Ghadimi, Milad Lankarany.

**Data curation:** Leon Amadeus Steiner, Luka Milosevic.

**Formal analysis:** Alireza Ghadimi, Leon Amadeus Steiner.

**Funding acquisition:** Milos R. Popovic, Milad Lankarany.

**Investigation:** Luka Milosevic, Milad Lankarany.

**Methodology:** Alireza Ghadimi, Milad Lankarany.

**Project administration:** Milos R. Popovic, Luka Milosevic, Milad Lankarany.

**Resources:** Milad Lankarany.

**Software:** Alireza Ghadimi.

**Supervision:** Milos R. Popovic, Luka Milosevic, Milad Lankarany.

**Validation:** Alireza Ghadimi, Leon Amadeus Steiner, Luka Milosevic.

**Visualization:** Alireza Ghadimi.

**Writing – original draft:** Alireza Ghadimi, Milad Lankarany.

**Writing – review & editing:** Leon Amadeus Steiner, Milos R. Popovic, Luka Milosevic, Milad Lankarany.

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
