## [Decision Letter · Decision Letter 0]

15 May 2022

PONE-D-22-01502Inferring Stimulation Induced Short-term Synaptic Plasticity Dynamics Using Novel Dual Optimization AlgorithmPLOS ONE

Dear Dr. Lankarany,

Thank you for submitting your manuscript to PLOS ONE. After careful consideration, we feel that it has merit but does not fully meet PLOS ONE’s publication criteria as it currently stands. Therefore, we invite you to submit a revised version of the manuscript that addresses the points raised during the review process.

We look forward to receiving your revised manuscript.

Kind regards,

Jun Ma, Dr.

Academic Editor

PLOS ONE

Journal Requirements:

This study was supported by the following grants and scholarships:

AGEWELL, UofT FASE Graduate Student and Postdoctoral Award in Technology and Aging (A.G.);

Deutsche Forschungsgemeinschaft (DFG, German Research Fundation) Project ID 424778381 TRR 295 (L.A.S.);

Junior Clinician Scientist Program of the Berlin Institute of Health (L.A.S.);

German Academic Exchange Service, DAAD (L.A.S.);

Brain Canada Foundation (L.M.);

Walter and Maria Schroeder Foundation (L.M.)

Natural Sciences and Engineering Research Council of Canada, RGPIN-2020-05868 (M.L)

We do not report any original data. We tested the performance of our proposed algorithm on a previously published data (see reference 7 in the submitted manuscript).

Reviewers' comments:

Reviewer's Responses to Questions

**Comments to the Author**

1. Does the manuscript provide a valid rationale for the proposed study, with clearly identified and justified research questions?

Reviewer #1: Yes

2. Is the protocol technically sound and planned in a manner that will lead to a meaningful outcome and allow testing the stated hypotheses?

Reviewer #1: Yes

3. Is the methodology feasible and described in sufficient detail to allow the work to be replicable?

Reviewer #1: Yes

4. Have the authors described where all data underlying the findings will be made available when the study is complete?

Reviewer #1: Yes

5. Is the manuscript presented in an intelligible fashion and written in standard English?

Reviewer #1: Yes

6. Review Comments to the Author

You may also provide optional suggestions and comments to authors that they might find helpful in planning their study.

Reviewer #1: The paper is interesting. It is a great deal to fit a model to experimental data. I would suggest to describe this in more details in the Introduction. There were several books published in this field previously. See for instance:

1. Chaos and Its Reconstructions, New York: Nova Science Publishers; 2003. (Eds). G. Gouesbet, S. Meunier-Guttin-Cluzel, O. Ménard.

2. Bezruchko BP, Smirnov DA. Extracting Knowledge From Time Series: (An In- troduction to Nonlinear Empirical Modeling. New York: Springer; 2010.

There could be some newer also.

Please, rebuild all figures. The figures are of very bad quality and some of them even cannot be read.

It is hard to understand why the Results section is placed before the Methods section, if this is journal's policy I can only pity all of us.

What do the strikeout terms in Eq. (15,16,22,23) mean?

Please, report details of both optimization algorithms used: Trust-Region optimization methods and Nelder-Mead algorithm.

Some programming functions like "argmax" and "argmin" appear in the text, but they are not suitable there. They should be explained or, better, reformulated.

In Eq. 28 both * and × were used, for what?

I did not found how many series and individuals were evaluated. Maybe, I was inattentive, but this affects the robustness of outcomes. What is the variability of fitting results between different series and individuals?

7. PLOS authors have the option to publish the peer review history of their article (what does this mean?). If published, this will include your full peer review and any attached files.

Reviewer #1: **Yes: **Ilya V. Sysoev

---

## [Author Response · Author response to Decision Letter 0]

4 Jul 2022

We thank the reviewer for the constructive feedback. These comments are reproduced below in blue together with our responses in black. Major changes to the manuscript text are indicated in red. We have striven to address all concerns. We believe that our paper has been significantly improved by addressing the reviewer’s comments and we hope that it will now be considered suitable for publication in the prestigious Plos One Journal.

Reviewer #1: 

The paper is interesting. It is a great deal to fit a model to experimental data.

Re: Thank you. We appreciate your understanding about the importance of this methodology for neural data. 

• I would suggest to describe this in more details in the Introduction. There were several books published in this field previously. See for instance:

1. Chaos and Its Reconstructions, New York: Nova Science Publishers; 2003. (Eds). G. Gouesbet, S. Meunier-Guttin-Cluzel, O. Ménard.

2. Bezruchko BP, Smirnov DA. Extracting Knowledge From Time Series: (An Introduction to Nonlinear Empirical Modeling. New York: Springer; 2010.

Re: We have added these references in Introduction. Thanks for your suggestions.

• Please, rebuild all figures. The figures are of very bad quality and some of them even cannot be read.

Re: The figures that were integrated in the PDF has reduced quality. The high-quality figures were attached to the document (in the "other" file). We request the reviewer to kindly consider that we used the figures within the text in the revised manuscript to enhance its reading, but it was the journal policy to keep figures away from the main body text. We will make sure all figures, with original high qualities, be used in the final structure for publication. This is why we provided "other" document including all figures within the text. We hope that the reviewer can access to this document to endorse the quality of original figures.

• It is hard to understand why the Results section is placed before the Methods section, if this is journal's policy I can only pity all of us.

Re: Yes, it was the journal policy to have the results prior to the method section. 

• What do the strikeout terms in Eq. (15,16,22,23) mean?

Re: Required explanations were added to the revised manuscript. 

• Please, report details of both optimization algorithms used: Trust-Region optimization methods and Nelder-Mead algorithm.

Re: These detials were added to the revised manuscript.

• Some programming functions like "argmax" and "argmin" appear in the text, but they are not suitable there. They should be explained or, better, reformulated.

Re: Required explanations were added to the revised manuscript.

• In Eq. 28 both * and × were used, for what?

Re: It was a typo and it is corrected in the revised manuscript.

• I did not found how many series and individuals were evaluated. Maybe, I was inattentive, but this affects the robustness of outcomes. What is the variability of fitting results between different series and individuals?

Re: Good observation. The data used in this work was from 1 individual, we used this data as for a proof of principal. Thank you.

---

## [Decision Letter · Decision Letter 1]

14 Jul 2022

PONE-D-22-01502R1Inferring Stimulation Induced Short-term Synaptic Plasticity Dynamics Using Novel Dual Optimization AlgorithmPLOS ONE

Dear Dr. Lankarany,

Thank you for submitting your manuscript to PLOS ONE. After careful consideration, we feel that it has merit but does not fully meet PLOS ONE’s publication criteria as it currently stands. Therefore, we invite you to submit a revised version of the manuscript that addresses the points raised during the review process.

We look forward to receiving your revised manuscript.

Kind regards,

Jun Ma, Dr.

Academic Editor

PLOS ONE

Journal Requirements:

Reviewers' comments:

Reviewer's Responses to Questions

**Comments to the Author**

1. If the authors have adequately addressed your comments raised in a previous round of review and you feel that this manuscript is now acceptable for publication, you may indicate that here to bypass the “Comments to the Author” section, enter your conflict of interest statement in the “Confidential to Editor” section, and submit your "Accept" recommendation.

Reviewer #1: All comments have been addressed

2. Is the manuscript technically sound, and do the data support the conclusions?

Reviewer #1: Yes

3. Has the statistical analysis been performed appropriately and rigorously? 

Reviewer #1: N/A

4. Have the authors made all data underlying the findings in their manuscript fully available?

Reviewer #1: Yes

5. Is the manuscript presented in an intelligible fashion and written in standard English?

Reviewer #1: Yes

6. Review Comments to the Author

Reviewer #1: Most raised issues were reasonably addressed. I have to comment two new points which appeared in the revised manuscript and were not present in the original paper.

1. The only rat was considered. So, the model reconstruction was done only just as a proof of concept. In any case I do support this research. But this should be clearly stated in both abstract and Introduction.

2. The second derivative was calculated from the data. Experimental data are noisy and all numerical approaches are very sensitive to measurement noise. Please, comment what was done to minimize the effect of the noise (e.g. Savitzky–Golay filter). Also, please, take into account that in some recent papers the models integrated over time are preferred for identification, since this reduces number of differentiations, see e.g.

[1] Mishchenko et al. Identification of Phase-Locked Loop System from Its Experimental Time Series. IEEE Transactions on Circuits and Systems II: Express Briefs, 2022. 69(3):854-858 doi: 10.1109/TCSII.2021.3122892

[2] Sysoev & Bezruchko. Noise robust approach to reconstruction of van der Pol-like oscillators and its application to Granger causality. Chaos 31, 083118 (2021); doi: 10.1063/5.0056901

Both these papers actually deal with some neuron-like models though this is not clear from the title. Please, study, whether reconstruction of integrated over time equations is theoretically possible for the considered system.

7. PLOS authors have the option to publish the peer review history of their article (what does this mean?). If published, this will include your full peer review and any attached files.

Reviewer #1: No

---

## [Author Response · Author response to Decision Letter 1]

8 Aug 2022

We thank the reviewer for the constructive feedback. We hope that it will now be considered suitable for publication in the prestigious Plos One Journal.

Reviewer #1: 

Most raised issues were reasonably addressed. I have to comment two new points which appeared in the revised manuscript and were not present in the original paper.

1. The only rat was considered. So, the model reconstruction was done only just as a proof of concept. In any case I do support this research. But this should be clearly stated in both abstract and Introduction.

Re: Thank you. Your point is very true. We added this point in both the Abstract and the Introduction. Please see the revised version.

2. The second derivative was calculated from the data. Experimental data are noisy and all numerical approaches are very sensitive to measurement noise. Please, comment what was done to minimize the effect of the noise (e.g. Savitzky–Golay filter). Also, please, take into account that in some recent papers the models integrated over time are preferred for identification, since this reduces number of differentiations, see e.g.

[1] Mishchenko et al. Identification of Phase-Locked Loop System from Its Experimental Time Series. IEEE Transactions on Circuits and Systems II: Express Briefs, 2022. 69(3):854-858 doi: 10.1109/TCSII.2021.3122892

[2] Sysoev & Bezruchko. Noise robust approach to reconstruction of van der Pol-like oscillators and its application to Granger causality. Chaos 31, 083118 (2021); doi: 10.1063/5.0056901

Both these papers actually deal with some neuron-like models though this is not clear from the title. Please, study, whether reconstruction of integrated over time equations is theoretically possible for the considered system.

Re: Your point is reasonable, this is why we added a new sub-section to our Methodology part, sub-section 2.6.1, to address your point. We mainly put our focus to provide a context for the suggested references as they are suitable to deal with noise using numerical integration (rather than the differentiation). As well to justify why our algorithm can be reasonably robust to experimental noise, we considered the steady-state estimation of the TM model, which can be calculated from the recorded PSC over time (average). Please note that since the scope of this paper and proposed algorithm were not on the exploration of the impact of noise, we think that this sub-section can address the importance of noise in experimental data without diverging the focus of the paper. Thank you.

---

## [Decision Letter · Decision Letter 2]

15 Aug 2022

Inferring Stimulation Induced Short-term Synaptic Plasticity Dynamics Using Novel Dual Optimization Algorithm

PONE-D-22-01502R2

Dear Dr. Lankarany,

We’re pleased to inform you that your manuscript has been judged scientifically suitable for publication and will be formally accepted for publication once it meets all outstanding technical requirements.

Kind regards,

Jun Ma, Dr.

Academic Editor

PLOS ONE

Reviewer #1: All comments have been addressed

---

## [Editor Report · Acceptance letter]

7 Sep 2022

PONE-D-22-01502R2 

Inferring Stimulation Induced Short-term Synaptic Plasticity Dynamics Using Novel Dual Optimization Algorithm 

Dear Dr. Lankarany:

I'm pleased to inform you that your manuscript has been deemed suitable for publication in PLOS ONE. Congratulations! Your manuscript is now with our production department. 

Kind regards, 

on behalf of

Dr. and Pro. Jun Ma 

Academic Editor

PLOS ONE